# Motor Fluctuations Development Is Associated with Non-Motor Symptoms Burden Progression in Parkinson’s Disease Patients: A 2-Year Follow-Up Study

**DOI:** 10.3390/diagnostics12051147

**Published:** 2022-05-05

**Authors:** Diego Santos-García, Teresa de Deus Fonticoba, Carlos Cores Bartolomé, Maria J. Feal Painceiras, Ester Suárez Castro, Héctor Canfield, Cristina Martínez Miró, Silvia Jesús, Miquel Aguilar, Pau Pastor, Lluís Planellas, Marina Cosgaya, Juan García Caldentey, Nuria Caballol, Ines Legarda, Jorge Hernández-Vara, Iria Cabo, Lydia López Manzanares, Isabel González Aramburu, Maria A. Ávila Rivera, Víctor Gómez Mayordomo, Víctor Nogueira, Víctor Puente, Julio Dotor García-Soto, Carmen Borrué, Berta Solano Vila, María Álvarez Sauco, Lydia Vela, Sonia Escalante, Esther Cubo, Francisco Carrillo Padilla, Juan C. Martínez Castrillo, Pilar Sánchez Alonso, Maria G. Alonso Losada, Nuria López Ariztegui, Itziar Gastón, Jaime Kulisevsky, Marta Blázquez Estrada, Manuel Seijo, Javier Rúiz Martínez, Caridad Valero, Mónica Kurtis, Oriol de Fábregues, Jessica González Ardura, Ruben Alonso Redondo, Carlos Ordás, Luis M. López Díaz, Darrian McAfee, Pablo Martinez-Martin, Pablo Mir

**Affiliations:** 1Department of Neurology, Hospital Universitario de A Coruña (HUAC), Complejo Hospitalario Universitario de A Coruña (CHUAC), C/As Xubias 84, 15006 A Coruña, Spain; 2CHUF, Complejo Hospitalario Universitario de Ferrol, 15006 A Coruña, Spain; 3Unidad de Trastornos del Movimiento, Servicio de Neurología y Neurofisiología Clínica, Instituto de Biomedicina de Sevilla, Hospital Universitario Virgen del Rocío/CSIC/Universidad de Sevilla, 41013 Seville, Spain; 4CIBERNED (Centro de Investigación Biomédica en Red Enfermedades Neurodegenerativas), 28031 Madrid, Spain; 5Hospital Universitari Mutua de Terrassa, 08221 Terrassa, Barcelona, Spain; 6Clínica del Pilar, 08006 Barcelona, Spain; 7Hospital Clínic de Barcelona, 08036 Barcelona, Spain; 8Centro Neurológico Oms 42, 07003 Palma de Mallorca, Spain; 9Consorci Sanitari Integral, Hospital Moisés Broggi, 08970 Sant Joan Despí, Barcelona, Spain; 10Hospital Universitario Son Espases, 07120 Palma de Mallorca, Spain; 11Hospital Universitario Vall d’Hebron, 08035 Barcelona, Spain; 12Complejo Hospitalario Universitario de Pontevedra (CHOP), 36071 Pontevedra, Spain; 13Hospital Universitario La Princesa, 28006 Madrid, Spain; 14Hospital Universitario Marqués de Valdecilla, 39008 Santander, Spain; 15Consorci Sanitari Integral, Hospital General de L’Hospitalet, L’Hospitalet de Llobregat, 08906 Barcelona, Spain; 16Hospital Universitario Clínico San Carlos, 28040 Madrid, Spain; 17Hospital Da Costa, 27880 Burela, Lugo, Spain; 18Hospital del Mar, 08003 Barcelona, Spain; 19Hospital Universitario Virgen Macarena, 41009 Sevilla, Spain; 20Hospital Infanta Sofía, 28703 Madrid, Spain; 21Institut d’Assistència Sanitària (IAS)—Institut Català de la Salut, 17190 Girona, Spain; 22Hospital General Universitario de Elche, 03203 Elche, Spain; 23Fundación Hospital de Alcorcón, 28922 Madrid, Spain; 24Hospital de Tortosa Verge de la Cinta (HTVC), Tortosa, 43500 Tarragona, Spain; 25Complejo Asistencial Universitario de Burgos, 09006 Burgos, Spain; 26Hospital Universitario de Canarias, 38320 San Cristóbal de la Laguna, Santa Cruz de Tenerife, Spain; 27Hospital Universitario Ramón y Cajal, IRYCIS, 28034 Madrid, Spain; 28Hospital Universitario Puerta de Hierro, 28222 Madrid, Spain; 29Hospital Álvaro Cunqueiro, Complejo Hospitalario Universitario de Vigo (CHUVI), 36213 Vigo, Spain; 30Complejo Hospitalario de Toledo, 45004 Toledo, Spain; 31Complejo Hospitalario de Navarra, 31008 Pamplona, Spain; 32Hospital de Sant Pau, 08041 Barcelona, Spain; 33Hospital Universitario Central de Asturias, 33011 Oviedo, Spain; 34Hospital Universitario Donostia, 20014 San Sebastián, Spain; 35Hospital Arnau de Vilanova, 46015 Valencia, Spain; 36Hospital Ruber Internacional, 28034 Madrid, Spain; 37Hospital de Cabueñes, 33394 Gijón, Spain; 38Hospital Universitario Lucus Augusti (HULA), 27003 Lugo, Spain; 39Hospital Rey Juan Carlos, 28933 Madrid, Spain; 40Complejo Hospitalario Universitario de Orense (CHUO), 32005 Orense, Spain; 41University of Maryland School of Medicine, Baltimore, MD 21201, USA

**Keywords:** burden, follow-up, non-motor symptoms, motor fluctuations, Parkinson’s disease

## Abstract

**Objective:** The aim of the present study was to analyze the progression of non-motor symptoms (NMS) burden in Parkinson’s disease (PD) patients regarding the development of motor fluctuations (MF). **Methods:** PD patients without MF at baseline, who were recruited from January 2016 to November 2017 (V0) and evaluated again at a 2-year follow-up (V2) from 35 centers of Spain from the COPPADIS cohort, were included in this analysis. MF development at V2 was defined as a score ≥ 1 in the item-39 of the UPDRS-Part IV, whereas NMS burden was defined according to the Non-motor Symptoms Scale (NMSS) total score. **Results:** Three hundred and thirty PD patients (62.67 ± 8.7 years old; 58.8% males) were included. From V0 to V2, 27.6% of the patients developed MF. The mean NMSS total score at baseline was higher in those patients who developed MF after the 2-year follow-up (46.34 ± 36.48 vs. 34.3 ± 29.07; *p* = 0.001). A greater increase in the NMSS total score from V0 to V2 was observed in patients who developed MF (+16.07 ± 37.37) compared to those who did not develop MF (+6.2 ± 25.8) (*p* = 0.021). Development of MF after a 2-year follow-up was associated with an increase in the NMSS total score (β = 0.128; *p* = 0.046) after adjustment to age, gender, years from symptoms onset, levodopa equivalent daily dose (LEDD) and the NMSS total score at baseline, and the change in LEDD from V0 to V2. **Conclusions:** In PD patients, the development of MF is associated with a greater increase in the NMS burden after a 2-year follow-up.

## 1. Introduction

Parkinson’s disease (PD) is a progressive neurodegenerative disorder causing motor and non-motor symptoms (NMS) that result in disability, loss of patient autonomy, and diminished quality of life (QoL) [1]. From a pathophysiological point of view, motor symptoms in PD are attributed to the degeneration of the dopaminergic nigrostriatal system [2]. Nevertheless, increasing evidence has shown that PD is a multisystem disorder characterized also by the degeneration of the mesocortical dopaminergic system, the noradrenergic system of the locus coeruleus, the serotonergic system of the dorsal raphe nuclei, and the cholinergic system of the nucleus basalis of Meynert, as well as the histaminergic, peptidergic, and olfactory-related systems [3]. This explains the complexity in management of NMS in PD and why many therapeutic strategies are based on correcting the deficit of neurotransmitters other than dopamine [4]. However, NMS can be related to dopamine as well. Increasing dopamine activity not only in the striatum but also in other areas of the brain could improve some NMS such as attention, executive functions, apathy, depression, anxiety, restless legs and periodic limb movements, urinary urgency, nocturia, dribbling of saliva, constipation, pain, or fatigue [5,6,7,8,9]. Moreover, NMS can be related to dopamine changes in brain and blood [10]. Thus, some patients can suffer from non-motor fluctuations (NMF) (i.e., NMS that fluctuate during the day) [11] or can experience motor fluctuations (MF) with the development of NMS during the OFF episodes (e.g., pain associated with dystonia) [12]. The close connection of NMF and MF strongly suggests that the strategies used to treat motor complications—namely, continuous dopaminergic stimulation—also apply for the therapy of NMF. Thus, a dopaminergic treatment reducing the daily OFF time can improve some NMS [9,13,14] or even the global NMS burden [15,16]. In line with this, we demonstrated recently in a cross-sectional study conducted in Spain that MF are frequent and associated with a greater NMS burden even during the first 5 years of disease duration [17]. This is of great importance because NMS burden is associated with a worse QoL and is also an independent predictor of clinically significant QoL impairment in PD [18,19]. 

In this context, we hypothesized that PD patients who develop MF in the short-term will increase their NMS burden compared with those patients who do not. Understanding this potential association is of interest because, in clinical practice, to detect MF is an essential point for the application of management strategies in PD [20]. The aim of the present study was to analyze the progression of NMS burden in PD patients from a Spanish cohort regarding the development of MF after a 2-year follow-up. Moreover, the change in health-related quality of life (HR-QoL) and global QoL (GQoL) was analyzed as well.

## 2. Material and Methods

PD patients without MF at baseline, who were recruited from 35 centers of Spain from the COPPADIS cohort [21] from January 2016 to November 2017 and evaluated again at 2-year follow-up, were included in the study. Methodology about COPPADIS-2015 study can be consulted in https://bmcneurol.biomedcentral.com/articles/10.1186/s12883-016-0548-9 accessed on 25 February 2016 [22]. This is a multicenter, observational, longitudinal-prospective, 5-year follow-up study designed to analyze disease progression in a Spanish population of PD patients. All patients included were diagnosed according to UK PD Brain Bank criteria [22].

In PD subjects, information on sociodemographic aspects, factors related to PD, comorbidity, and treatment was collected at baseline (visit V0) and at 2 years ± 1 month (visit V2). V0 and V2 evaluations included motor assessment (Hoenh & Yahr [H&Y], Unified Parkinson’s Disease Rating Scale [UPDRS] part III and part IV, Freezing of Gait Questionnaire [FOGQ]), NMS (Non-Motor Symptoms Scale [NMSS], Parkinson’s Disease Sleep Scale [PDSS], Visual Analog Scale-Pain [VAS-Pain], Visual Analog Fatigue Scale [VAFS]), cognition (PD-CRS), mood and neuropsychiatric symptoms (Beck Depression Inventory-II [BDI-II], Neuropsychiatric Inventory [NPI], Questionnaire for Impulsive-Compulsive Disorders in Parkinson’s Disease-Rating Scale [QUIP-RS]), disability (Schwab & England Activities of Daily Living Scale [ADLS]), and QoL (the 39-item Parkinson’s disease Questionnaire [PDQ-39], the EUROHIS-QOL 8-item index [EUROHIS-QOL8]) [22]. In all the scales/questionnaires, a higher score indicates a more severe affectation except for PD-CRS, PDSS, ADLS, and EUROHIS-QOL8, where it is opposite. 

MF were defined according to the Unified Parkinson’s Disease Rating Scale–Part IV (UPDRS-IV) [23]. Patients with a score = 0 on item-39 of the UPDRS-IV (UPDRS-IV-39) were considered as without MF whereas those with a UPDRS-IV-39 score ≥ 1 were defined as with MF. For this study, patients from the COPPADIS cohort who presented with MF (i.e., UPDRS-IV-39 ≥ 1) at baseline were excluded. In patients with MF, the motor assessment was made during the OFF state (without medication in the last 12 h) and during the ON state. On the other hand, the assessment was only performed without medication in patients without MF. Other data about motor complications were obtained from the UPDRS-IV. 

The NMS burden was defined according to the NMSS total score [24]. The NMSS includes 30 items, each with a different non-motor symptom. The symptoms refer to the 4 weeks prior to assessment. The total score for each item is the result of multiplying the frequency (0, never; 1, rarely; 2, often; 3, frequent; 4, very often) × severity (1, mild; 2, moderate; 3, severe) and will vary from 0 to 12 points. The scale score ranges from 0 to 360 points. The items are grouped into 9 different domains: (1) Cardiovascular (items 1 and 2; score, 0 to 24); (2) Sleep/fatigue (items 3, 4, 5, and 6; score, 0 to 48); (3) Mood/apathy (items 7, 8, 9, 10, 11, and 12; score, 0 to 72); (4) Perceptual problems/hallucinations (items 13, 14, and 15; score, 0 to 36); (5) Attention/memory (items 16, 17, and 18; score, 0 to 36); (6) Gastrointestinal symptoms (items 19, 20, and 21; score 0 to 36); (7) Urinary symptoms (items 22, 23, and 24; score, 0 to 36); (8) Sexual dysfunction (items 25 and 26; score 0 to 24); (9) Miscellaneous (items 27, 28, 29, and 30; score, 0 to 48). Regarding the NMS burden, different groups were defined: mild (NMSS 1–20); moderate (NMSS 21–40); severe (NMSS 41–70); very severe (NMSS > 70) [25].

The PDQ-39 [26] and EUROHIS-QOL8 [27] were used to assess the HRQoL and GQoL, respectively. The PDQ-39 includes 39 items grouped into 8 domains: (1) Mobility (items 1 to 10); (2) Activities of daily living (ADL) (items 11 to 16); (3) Emotional well-being (items 17 to 22); (4) Stigma (items 23 to 26); (5) Social support (items 27 to 29); (6) Cognition (items 30 to 33); (7) Communication (items 34 to 36); (8) Pain and discomfort (items 37 to 39). For each item, the score may range from 0 (never) to 4 (always). The symptoms refer to the 4 weeks prior to assessment. Domain total scores are expressed as a percentage of the corresponding maximum possible score and a Summary Index is obtained as average of the domain scores. The EUROHIS-QOL8 is an 8-item GQoL questionnaire (quality of life, health status, energy, autonomy for ADL, self-esteem, social relationships, economic capacity, and habitat) derived from the WHOQOL-BREF. For each item, the score ranges from 0 (not at all) to 5 (completely). The total score is expressed as the mean of the individual scores. A higher score indicates a better QoL. 

## 3. Data Analysis

Data were processed using SPSS 20.0 for Windows. Only PD patients from the COPPADIS cohort with data of the UPDRS-IV and NMSS total score collected at both visits, V0 and V2, were included in the analysis. For comparisons between patients with vs. without MF at V2, the Student’s *t*-test, Mann–Whitney U test, Chi-square test, or Fisher test were used as appropriate (distribution for variables was verified by one-sample Kolmogorov-Smirnov test). Spearman’s or Pearson’s correlation coefficient, as appropriate, were used for analyzing the relationship between the change from V0 to V2 in continuous variables (NMSS, PDQ-39SI, EUROHIS-QOL8). Correlations were considered weak for coefficient values ≤ 0.29, moderate for values between 0.30 and 0.59, and strong for values ≥0.60. Marginal homogeneity tests were applied for comparing the frequency distribution of groups (NMS burden; from mild to very severe) between V0 and V2.

General linear model (GLM) repeated measure was used to test whether the mean differences of the total score and each domain of the NMSS, PDQ-39SI, and EUROHIS-QOL8 between the two visits (V0 and V2) were significant. The Bonferroni method was used as a post-hoc test after ANOVA. Cohen’s d formula was applied for measuring the effect size; it was considered as follows: <0.2—Negligible; 0.2–0.49—Small; 0.50–0.79—Moderate; ≥0.80—Large. Age, gender, years from symptoms onset, H&Y stage, levodopa equivalent daily dose (LEDD) and the NMSS total score at baseline, and the change in LEDD from V0 to V2 were included as covariates in the model. The total score of each scale at V0 (NMSS, PDQ-39SI, and EUROHIS-QOL8) was included as covariate for the analysis of their domains. 

With the aim to investigate if the development of MF from V0 to V2 was an independent factor associated with an increase in the NMS burden and impairment in the QoL, linear regression models with the change from V0 to V2 in the total score of the NMSS, PDQ-39SI, and EUROHIS-QOL8 (these variables as dependent variable in each model) were conducted. In all cases, the analysis was adjusted to age, gender, years from symptoms onset, H&Y stage, LEDD and the NMSS total score at baseline, and the change in LEDD from V0 to V2. The *p*-value was considered significant when it was <0.05.

## 4. Standard Protocol Approvals, Registrations, and Patient Consents

For this study, we received approval from the *Comité de Ética de la Investigación Clínica de Galicia* in Spain (2014/534; 02/DEC/2014). Written informed consent was obtained from all participants in this study. COPPADIS-2015 was classified by the AEMPS (*Agencia Española del Medicamento y Productos Sanitarios*) as a Postauthorization Prospective Follow-up study with the code COH-PAK-2014-01.

## 5. Data Availability

The data that support the findings of this study are available on request from the corresponding author. The data are not publicly available due to privacy or ethical restrictions.

## 6. Results

Three hundred and thirty PD patients (62.67 ± 8.7 years old; 58.8% males) without MF at baseline were included. From V0 to V2, 27.6% of the patients (91/330) developed MF. In the group of patients with MF at V2, OFF episodes were predictable in 89% of the cases and unpredictable in 15.4%; early morning dystonia was reported by 25.3% of the patients; and the proportion of the waking day during the OFF state was 82.4% from 1 to 25%, 16.5% from 26 to 50%, and only 1 patient with >50%. Thirty-six out of 91 patients who developed MF (39.6%) presented dyskinesia as well, being disabling in 15 patients (15/36; 41.7%). 

Compared with those patients who did not develop MF from V0 to V2, at V0, patients who presented with MF at V2 were younger (60.75 ± 9.06 vs. 63.41 ± 8.46 years old; *p* = 0.012), had a longer disease duration (5.36 ± 3.51 vs. 3.65 ± 3.09 years from symptoms onset; *p* < 0.0001); were receiving more dopaminergic medication; and had a worse status in terms of motor symptoms, NMS, QoL, and autonomy for ADL (Table 1). The mean NMSS total score at baseline was higher in those patients who developed MF after the 2-year follow-up than in those who did not develop MF (46.34 ± 36.48 vs. 34.3 ± 29.07; *p* = 0.001) (Table 1 and Figure 1). At V0, the frequency of severe and very severe NMS burden was higher in those patients who developed MF at V2 compared with those who did not (27.5% vs. 18% and 18.7% vs. 12.1%, respectively; *p* = 0.011) (Figure 2).

A greater increase in the NMSS total score from V0 to V2 was observed in those patients who developed MF at V2 (+16.07 ± 37.37) compared with those who did not develop MF (+6.2 ± 25.8) (*p* = 0.021) (Table 1 and Figure 1). Two-hundred and two out of 330 patients (64.2%) presented at V2 a NMSS total score higher than at V0, but no differences between patients who developed MF vs. those who did not develop MF after the 2-year follow-up were observed (68.1% vs. 62.1%; *p* = 0.218). However, after the 2-year follow-up, the frequency of severe and very severe NMS burden was significantly higher in the group who developed MF (*p* < 0.0001) (Figure 2). Applying GLM repeated measure and after adjustment to covariates (age, gender, years from symptoms onset, H&Y stage, LEDD and the NMSS total score at baseline, and the change in LEDD from V0 to V2), a significantly greater increase (34.6% vs. 17.9%; *p* = 0.005) in the NMSS total score was observed in patients who developed MF at V2 (from 46.34 ± 36.48 to 62.37 ± 44.15; Cohen’s effect size = 0.57; *p* = 0.003) compared with those who did not develop MF (from 34.3 ± 29.07 vs. 40.5 ± 35.4; Cohen’s effect size = 0.33; *p* < 0.0001) (Table 2). An increase in the score of different domains from V0 to V2 was significant in both groups, with and without MF at V2, but there were no significant differences between them (Table 2 and Figure 1). Regarding QoL, the increase in the PDQ-39SI and decrease in EUROHIS-QOL8 total score indicating a QoL impairment between both visits, V0 and V2, was significantly greater in the group of patients who developed MF (PDQ-39SI, +35% vs. +26.5% (*p* = 0.002); EUROHIS-QOL8, −29.9% vs. −0.7% (*p* = 0.030)) (Table 2). By domain and after adjustment to covariates including the PSQ-39SI score at V0, the increase on the score of “pain and discomfort” domain in the group who developed MF at V2 (from 28.55 ± 20.01 to 32.87 ± 24.33; Cohen’s effect size = 0.30; *p* = 0.015) was significantly higher (*p* = 0.039) compared with patients who did not develop MF (from 20.65 ± 18.82 to 23.97 ± 22.12; Cohen’s d effect size = 0.16; *p* = 0.071) (Table 2). The mean score on all domains of the PDQ-39SI was the highest in patients who developed MF after the 2-year follow-up at V2 and the lowest in patients who did not develop MF, at V0 (Figure 3). A moderate correlation was observed between the change from V0 to V2 in the NMSS total score and the change in the PDQ-39SI in the whole cohort (N = 320; r = 0.402; *p* < 0.0001) and in both groups, patients with (N = 91; r = 0.328; *p* = 0.002) and without MF (N = 239; r = 0.433; <0.0001) at V2. However, the correlation between the change in the total score of the NMSS and the EUROHIS-QOL8 was only significant in patients who developed MF at V2 (N = 91; r = −0.277; *p* = 0.009) but not in patients who did not develop MF at V2 (N = 239; r = −0.111; *p* = 0.088). 

To develop MF after a 2-year follow-up was associated with an increase in the NMSS total score without controlling for other factors (β = 0.148; 95% CI, 2.69–16.98; *p* = 0.007) but also after adjustment to age, gender, years from symptoms onset, LEDD and the NMSS total score at baseline, and the change in LEDD from V0 to V2 as well (β = 0.128; 95% CI, 0.17–16.86; *p* = 0.046). However, when time on levodopa and the H&Y stage were included in the model as covariates, it was not significant (with time on levodopa therapy, *p* = 0.062; with H&Y, *p* = 0.167; both variables, *p* = 0.212). Development of MF was associated with an increase in the PDQ39SI from V0 to V2 (β = 0.135; 95% CI, 0.61–5.55; *p* = 0.015) but not with the change in the EUROHIS-QOL8 total score (*p* = 0.207). However, it was not significant after controlling for other covariates (age, gender, years from symptoms onset, LEDD and the PDQ-39SI at baseline, and the change in LEDD from V0 to V2) (*p* = 0.094). 

## 7. Discussion

The present study observes that MF are frequent in PD, appearing in a cohort of 330 patients with a mean of 4 years from symptoms onset in one of every 4 subjects after a 2-year follow-up, and also that they are related to NMS. Specifically, NMS burden was greater at baseline in PD patients who 2 years later developed MF, and the increase in the NMS burden after the 2-year follow-up was double in this group as well. Moreover, similar results were obtained in terms of QoL. Importantly, all patients at baseline were without MF and this is the first time that NMS burden progression is specifically analyzed regarding the development of MF in a PD cohort. 

MF are frequent in PD [28,29,30,31]. In the COPPADIS cohort, of 690 patients with a mean disease duration of 5.5 years (DS 4.37), 33.9% had MF [17]. This percentage was 18.1% in the subgroup of patients with ≤5 years of disease duration (N = 396), with a mean disease duration of 2.7 years (DS 1.5) from symptoms onset [17]. The frequency will depend in part on the methods used—from an interview to wearable tools—and how sensitive we can be to detect them [32]. Stocchi et al. analyzed wearing-off (WO) in 617 PD patients with a mean disease duration of 6.6 years (DS 4.6) and observed that neurologist identified the presence of WO with an interview in 56.9% of the patients, whereas the percentage was 67.3% when the self-rated 19-question Wearing-Off Questionnaire (WOQ-19) was administered [33]. Identifying fluctuations is important in PD patients for two reasons. Firstly, their presence is associated with a worse status in terms of motor, NMS, QoL, and autonomy for ADL [17]. Secondly, the therapeutic strategy is conditioned by their presence to the point that there are several drugs marketed with indication to be only for patients with MF [34].

MF (either early or advanced) can significantly add to the NMS burden in PD [35,36]. However, few studies specifically focused on the NMS prevalence in motor-fluctuating PD patients [17,37,38]. Recent data published of 1589 PD patients from the SYNAPSIS study support the high prevalence of NMS in PD patients with MF in real-life condition, thus reinforcing the need for assessing them for diagnostic accuracy and for delivering holistic care [37]. Using the NMSS, we previously observed a greater NMS burden in the group with MF in a cross-sectional study conducted in PD patients from the COPPADIS cohort. In particular, 28 out of the 30 NMS included in the NMSS were significantly more frequent in patients with MF compared with those who did not have MF, and the mean score of all domains of the NMSS except urinary symptoms and sexual dysfunction was significantly higher in the group with MF [17]. Watanabe et al. recently explored the changes in NMS and QoL during 52 weeks in 996 Japanese PD patients exhibiting MF using the Movement Disorder Society Unified PD Rating Scale (MDS-UPDRS) Part I and the 8-item PD Questionnaire (PDQ-8), respectively [38]. They detected that changes in MDS-UPDRS Part I scores were variable and related to changes in HRQoL and identified 3 separate groups: unchanged (63.8%); deteriorated (20.1%); improved (16.2%). However, very importantly, all patients included in this study had MF. To our knowledge, our study is the first one to prospectively analyze the change in the NMS burden in relation to the development of MF in PD patients who initially did not have MF. As we previously reported in this cohort [39], about 6 out of 10 patients increased the NMSS total score after a 2-year follow-up. Although there were no differences in the percentage between the two groups—patients who developed MF and patients who did not develop MF—a greater NMS burden increase was observed in the first group. We did not analyze specifically if NMS fluctuated (e.g., NMS-MDS [40]), but this finding would support the relationship between NMS and the presence of OFF episodes with an increase in NMS perception during the OFF episodes. Importantly, the effect of MF on NMS burden persisted after adjustment to some variables related to NMS in PD such as age, gender, disease duration, or even dopaminergic medication [35,41,42]. However, NMS in PD are related to motor stage as well [17,18,25,42], and after the inclusion of the H&Y stage in the model, the effects disappeared. The same happened when time on levodopa therapy was included as covariate in the model. It is well-known that both aspects are related to the development of MF [30,31]. A more advanced H&Y stage is related to a greater degree of denervation of the striatal nucleus with more sensitivity to the development of MF [43]. On the other hand, a longer time on levodopa could imply a longer disease duration but also more time exposed to certain causative mechanisms (presynaptic and postsynaptic changes and pharmacokinetic and pharmacodynamic factors) [30,31]. The data as a whole indicate that PD patients who will develop MF in the short-term are patients with a more advanced disease with a greater NMS burden and patients with an increased risk of developing more severe NMS burden. To detect NMS burden progression is relevant because it is associated with a worse QoL [18,28]; importantly, in this context, MF development was associated with a greater worsening of both HRQoL and GQoL in the present analysis. To reduce NMS burden in PD patients has been demonstrated to be associated with an improvement in QoL [14,15]. In summary, our findings reinforce the idea that there is a close relationship between motor and NMS and that dopaminergic treatment can be helpful in some cases [5,10]. 

The present study has some limitations. The sample size of the group of patients with MF at V2 was smaller (N = 91) compared with the group without MF (N = 231), and the information about NMS burden follow-up was recorded in 330 patients of 462 (71.4%) without MF at baseline from the COPPADIS cohort. This is a limitation observed in other prospective studies, with percentages ranging from 61.9% to 89.8% [39,42,44]. We used the NMSS to assess the NMS burden progression, but some studies suggest that a battery of separate NMS scales is more sensitive to change than the NMSS [45]. Our sample was not fully representative of the PD population due to inclusion and exclusion criteria (i.e., age limit, no dementia, no severe comorbidities, no second line therapies, etc.). For some variables, the information was not collected in all cases (the smallest sample size was for the change in NPI (N = 255) since it was covered by the caregiver and not all had a primary caregiver). On the contrary, the strengths of our study include a very thorough assessment, a prospective longitudinal follow-up design, and the extensive clinical and demographic information recorded.

In conclusion, we demonstrated for the first time in a prospective study that, in PD, the development of MF is associated with a greater NMS burden increase in the short-term. In practice, it is essential to detect MF early and ask about NMS, especially in patients with a greater disease severity and a longer time on levodopa. 

## Figures and Tables

**Figure 1 diagnostics-12-01147-f001:**
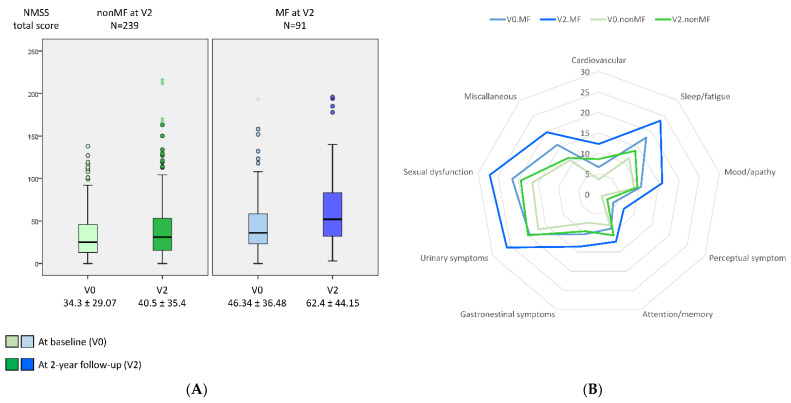
(**A**) NMSS total score (*y*-axis) at baseline (V0) and after a 2-year follow-up (V2) in PD patients who developed MF at V2 (MF at V2 (PD-MF_V2_); N = 91) and those patients who did not develop MF at V2 (nonMF at V2 (PD-nonMF_V2_); N = 239). NMSS total score at V0, PD-MF_V2_ vs. PD-nonMF_V2_, *p* = 0.001; NMSS total score at V2, PD-MF_V2_ vs. PD-nonMF_V2_, *p* < 0.0001; change in the NMSS total score from V0 to V2 in PD-MF_V2_, *p* < 0.0001; change in the NMSS total score from V0 to V2 in PD-nonMF_V2_, *p* < 0.0001; comparison between the change in the NMSS total score from V0 to V2 in PD-MF_V2_ vs. PD-nonMF_V2_, *p* = 0.021. Data are presented as box plots, with the box representing the median and the two middle quartiles (25–75%). (**B**) Mean score on each domain of the NMSS at V0 and at V2 in both groups, PD-MF_V2_ and PD-nonMF_V2_. At V0, the difference was significant between both groups in NMSS-1 (Cardiovascular) (*p* = 0.001), NMSS-2 (Sleep/fatigue) (*p* = 0.001), NMSS-4 (Perceptual symptoms) (*p* < 0.0001), and NMSS-9 (Miscellaneous) (*p* = 0.005). At V2, the difference was significant between both groups in all domains (*p* values from 0.024 to <0.0001) except in NMSS-5 (Attention/memory) (*p* = 0.364). *p* values were computed using the Kolmogorov–Smirnov, Mann–Whitney, and Wilcoxon tests. Mild outliers (O) are data points that are more extreme than Q1 − 1.5 * IQR or Q3 + 1.5 * IQR.

**Figure 2 diagnostics-12-01147-f002:**
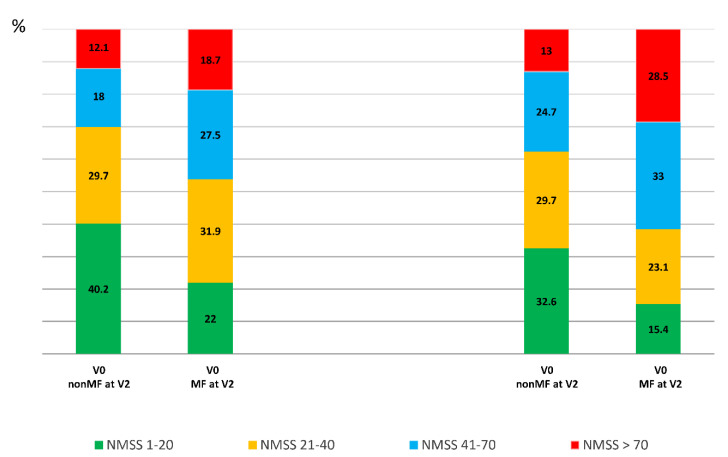
Frequency of patients with mild (NMSS 1–20), moderate (NMSS 21–40), severe (NMSS 41–70), and very severe (NMSS > 70) NMS burden at V0 and at V2 considering two groups: patients who developed MF at V2 (MF at V2 (PD-MF_V2_); N = 91) and those who did not developed MF at V2 (nonMF at V2 (PD-nonMF_V2_); N = 239). PD-nonMF_V2_ vs. PD-MF_V2_ at V0, *p* = 0.011; PD-nonMF_V2_ vs. PD-MF_V2_ at V2, *p* < 0.0001; change in PD-nonMF_V2_ from V0 to V2, *p* = 0.003; change in PD-MF_V2_ from V0 to V2, *p* = 0.001. *p* values were computed using the Chi-square and marginal homogeneity test.

**Figure 3 diagnostics-12-01147-f003:**
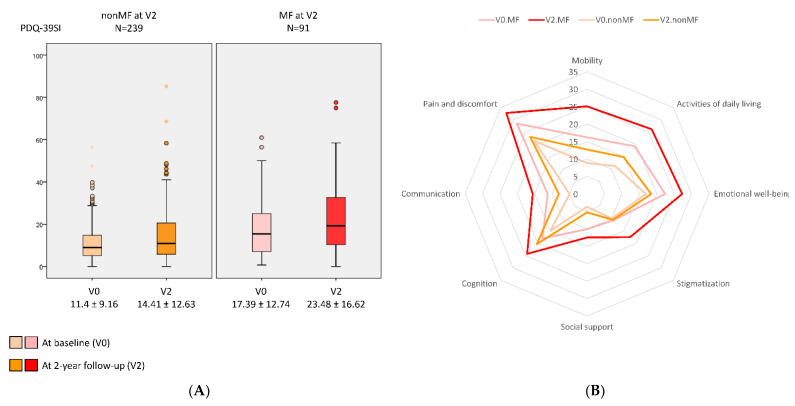
(**A**) QoL (PDQ-39SI) (*y*-axis) at baseline (V0) and after a 2-year follow-up (V2) (*x*-axis) in PD patients who developed MF at V2 (MF at V2 (PD-MFV2); N = 91) and those patients who did not developed MF at V2 (nonMF at V2 (PD-nonMFV2); N = 239). PDQ-39SI at V0, PD-MF_V2_ vs. PD-nonMF_V2_, *p* < 0.0001; PDQ-39SI at V2, PD-MF_V2_ vs. PD-nonMF_V2_, *p* < 0.0001; change in the PDQ-39SI from V0 to V2 in PD-MF_V2_, *p* < 0.0001; change in the PDQ-39SI from V0 to V2 in PD-nonMF_V2_, *p* < 0.0001; comparison between the change in the PDQ-39SI from V0 to V2 in PD-MF_V2_ vs. PD-nonMF_V2_, *p* = 0.005. (**B**) Mean score on each domain of the PDQ-39SI at V0 and at V2 in both groups, PD-MF_V2_ and PD-nonMF_V2_. At V0, the difference was significant between both groups in all domains (*p* values from 0.023 to <0.0001) except in PDQ-39SI-4 (Stigmatization) (*p* = 0.169) and PDQ-39SI-6 (Cognition) (*p* = 0.097). At V2, the difference was significant between both groups in all domains (*p* values from 0.005 to <0.0001) except in PDQ-39SI-6 (Cognition) (*p* = 0.319). PDQ-39 is expressed as a Summary Index (PDQ-39SI). Data are presented as box plots, with the box representing the median and the two middle quartiles (25–75%). *p* values were computed using the Kolmogorov–Smirnov, Mann–Whitney, and Wilcoxon tests. Mild outliers (O) are data points that are more extreme than Q1 − 1.5 * IQR or Q3 + 1.5 * IQR.

**Table 1 diagnostics-12-01147-t001:** Different PD-related variables in PD patients who developed MF at V2 (MF at V2; N = 91) compared with those patients who did not develop MF at V2 (nonMF at V2; N = 239).

	All Sample (N = 330)	nonMF at V2 (N = 239)	MF at V2 (N = 91)	*p*
Males (%)	58.8	59.8	56	0.308
**At V0**				
Age	62.67 ± 8.7	63.41 ± 8.46	60.75 ± 9.06	0.012
Years from symptoms onset	4.13 ± 3.3	3.65 ± 3.09	5.36 ± 3.51	<0.0001
Time on levodopa therapy (months)	18.99 ± 27.99	14.71 ± 24.37	29.65 ± 33.25	<0.0001
Daily dose of levodopa (mg/day)	231.85 ± 257.89	175.74 ± 216.46	379.62 ± 298.07	<0.0001
DA equivalent daily dose (mg/day)	152.77 ± 149.36	143.21 ± 148.24	177.96 ± 150.18	0.047
LEDD (mg/day)	437.71 ± 325.85	372.62 ± 283.4	609.1 ± 367.38	<0.0001
H&Y stage (OFF)				0.277
Stage from 1 to 3	99.7	100	98.8	
Stage from 4 to 5	0.3	0	1.2	
UPDRS-III (OFF)	18.9 ± 9.54	17.57 ± 8.81	22.4 ± 10.51	<0.0001
UPDRS-IV	0.71 ± 0.87	0.66 ± 0.79	0.86 ± 1.05	0.241
FOGQ	1.97 ± 3.13	1.56 ± 2.51	3.06 ± 4.19	<0.0001
Tremotic motor phenotype (%)	55.5	59	46.2	0.024
PD-CRS	92.93 ± 15.17	92.32 ± 15.39	94.51 ± 14.55	0.205
NMSS	37.62 ± 31.69	34.3 ± 29.07	46.34 ± 36.48	0.001
BDI-II	7.49 ± 6.63	6.98 ± 6.33	8.82 ± 7.22	0.037
PDSS	119.82 ± 23.36	122.41 ± 22.02	113.01 ± 25.45	<0.0001
QUIP-RS	3.68 ± 7.44	2.72 ± 5.94	6.42 ± 10.15	<0.0001
NPI	4.43 ± 6.62	4.22 ± 6.41	4.95 ± 7.13	0.381
VAS–PAIN	2.31 ± 2.8	2.22 ± 2.76	2.54 ± 2.91	0.363
VASF–physical	2.43 ± 2.57	2.27 ± 2.54	2.86 ± 2.61	0.050
VASF–mental	1.86 ± 2.45	1.75 ± 2.42	2.17 ± 2.51	0.084
PDQ-39SI	13.08 ± 10.59	11.44 ± 9.16	17.39 ± 12.74	<0.0001
EUROHIS-QOL8	3.87 ± 0.51	3.92 ± 0.5	3.74 ± 0.49	0.006
S&E-ADLS	91.12 ± 8.04	92.05 ± 7.24	88.68 ± 9.45	0.001
**Change at V2 (V2 vs. V0)**				
Daily dose of levodopa (mg/day)	+126.73 ± 190.01	+113.37 ± 186.82	+161.89 ± 208.83	0.021
DA equivalent daily dose (mg/day)	+13.35 ± 188.95	+6.32 ± 117.41	+31.85 ± 306.18	0.288
LEDD (mg/day)	+190.55 ± 278.38	+158.22 ± 222.42	+275.65 ± 377.62	0.008
UPDRS-III (OFF)	+3.5 ± 9.73	+2.11 ± 8.61	+7.01 ± 11.46	<0.0001
UPDRS-IV	+1.02 ± 2.09	+0.09 ± 1.03	+3.5 ± 2.18	<0.0001
FOGQ	+1.37 ± 3.63	+0.94 ± 3.18	+2.52 ± 4.44	0.001
PD-CRS	−0.9 ± 10.87	−0.79 ± 11.49	−1.19 ± 9.11	0.873
NMSS	+8.91 ± 29.77	+6.2 ± 25.8	+16.03 ± 37.37	0.021
BDI-II	+0.46 ± 7.16	+0.33 ± 7.2	+0.8 ± 7.07	0.463
PDSS	+0.61 ± 23.46	+1.05 ± 22.26	−0.55 ± 26.42	0.654
QUIP-RS	+0.79 ± 8.74	+0.93 ± 7.45	+0.42 ± 11.56	0.564
NPI	+0.4 ± 8.45	−0.23 ± 8.51	+1.89 ± 8.15	0.270
VAS–PAIN	+0.47 ± 3.15	+0.33 ± 3.13	+0.86 ± 3.17	0.109
VASF–physical	+0.57 ± 2.84	+0.42 ± 2.85	+0.94 ± 2.79	0.216
VASF–mental	+0.12 ± 2.75	−0.07 ± 2.61	+0.65 ± 3.04	0.062
PDQ-39SI	+3.85 ± 10.18	+3.01 ± 9.15	+6.09 ± 12.28	0.005
EUROHIS-QOL8	−0.05 ± 0.56	−0.03 ± 0.55	−0.12 ± 0.58	0.249
S&E-ADLS	−3.87 ± 9.73	−3.4 ± 9.35	−5.11 ± 10.62	0.177

Chi-square and Mann–Whitney–Wilcoxon tests were used. The results represent mean ± SD or %. ADLS, Schwab and England Activities of Daily Living Scale; BDI-II, Beck Depression Inventory-II; DA, dopamine agonist; FOGQ, Freezing Of Gait Questionnaire; LEDD, levodopa equivalent daily dose; N, number; NMSS, Non-motor Symptoms Scale; NPI, Neuropsychiatric Inventory; PD-CRS, Parkinson’s Disease Cognitive Rating Scale; PDSS, Parkinson’s Disease Sleep Scale; QUIP-RS, Questionnaire for Impulsive–Compulsive Disorders in Parkinson’s Disease-Rating Scale; TS, total score; UPDRS, Unified Parkinson’s Disease Rating Scale; VAFS, Visual Analog Fatigue Scale; VAS–Pain, Visual Analog Scale–Pain.

**Table 2 diagnostics-12-01147-t002:** Changes in non-motor symptoms and quality of life in PD patients who developed MF at V2 (MF at V2; N = 91) compared with those patients who did not develop MF at V2 (nonMF at V2; N = 239).

	nonMF at V2 V0	nonMF at V2 V2	Cohen’s Test	*p* ^a^	MF at V2 V0	MF at V2 V2	Cohen’s Test	*p* ^b^	*p* ^c^	*p* ^d^
NMSS	34.3 ± 29.07	40.5 ± 35.4	0.33	<0.0001	46.34 ± 36.48	62.37 ± 44.15	0.57	0.003	0.387	0.005
Cardiovascular	3.63 ± 7.35	8.66 ± 12.36	0.61	<0.0001	6.63 ± 10.22	12.36 ± 13.78	0.54	0.002	0.973	0.240
Sleep/fatigue	11.52 ± 13.03	13.91 ± 15.09	0.24	0.024	18.09 ± 16.8	23.53 ± 18.47	0.39	0.027	0.069	0.104
Mood/apathy	8.86 ± 13.56	9.68 ± 15.51	0.09	0.101	10.51 ± 16	15.82 ± 18.13	0.46	0.012	0.090	0.261
Perceptual symptoms	0.87 ± 3.48	2.41 ± 8.09	0.31	0.002	4.17 ± 8.84	7.11 ± 16	0.31	0.080	0.672	0.105
Attention/memory	8.05 ± 12.6	10.75 ± 16.24	0.27	0.002	8.93 ± 11.51	12.36 ± 15.78	0.32	0.062	0.736	0.175
Gastrointestinal symptoms	7.41 ± 10.67	9.61 ± 11.95	0.32	<0.0001	10.42 ± 14.81	13.64 ± 15.02	0.31	0.168	0.852	0.580
Urinary symptoms	17.45 ± 19.77	20.09 ± 21.92	0.21	0.035	19.63 ± 20.42	26 ± 23.95	0.43	0.042	0.923	0.532
Sexual dysfunction	16.58 ± 25.16	19.4 ± 26.65	0.14	0.119	21.55 ± 28.26	27.1 ± 27.55	0.24	0.128	0.980	0.082
Miscellaneous	10.96 ± 13.06	11.68 ± 12.83	0.07	0.945	15.84 ± 16.16	19.84 ± 16	0.34	<0.0001	0.058	0.060
PDQ-39SI	11.4 ± 9.16	14.41 ± 12.63	0.46	<0.0001	17.39 ± 12.74	23.48 ± 16.62	0.65	<0.0001	0.397	0.002
Mobility	8.84 ± 12.67	12.76 ± 15.86	0.42	<0.0001	16.2 ± 18.23	25.11 ± 22.48	0.69	<0.0001	0.034	N. A.
Activities of daily living	11.35 ± 13.19	14.98 ± 26.2	0.32	0.006	19.35 ± 19.34	26.2 ± 21.37	0.46	0.002	0.224	0.271
Emotional well-being	16.91 ± 17.25	18.41 ± 21.8	0.15	0.167	22.36 ± 20.74	27.38 ± 24.33	0.35	0.089	0.852	0.576
Stigmatization	10.03 ± 16.81	10.5 ± 18.63	0.05	0.477	10.7 ± 14.64	17.5 ± 22.91	0.46	0.002	0.032	N. A.
Social support	3.76 ± 10.59	5.3 ± 12.77	0.17	0.074	10.06 ± 17.04	12.59 ± 21.46	0.16	0.396	0.861	0.224
Cognition	14.94 ± 15.84	20.4 ± 18.11	0.49	<0.0001	18.05 ± 16.82	24.3 ± 22.43	0.42	0.027	0.833	0.424
Communication	5.01 ± 9.05	8.19 ± 14.68	0.33	<0.0001	11.34 ± 17.09	15.64 ± 19.42	0.26	0.056	0.574	0.387
Pain and discomfort	20.65 ± 18.82	23.97 ± 22.12	0.16	0.071	28.55 ± 20.01	32.87 ± 24.33	0.30	0.015	0.432	0.039
EUROHIS-QOL8	3.92 ± 0.5	3.89 ± 0.57	−0.07	0.699	3.74 ± 0.49	2.62 ± 0.54	−0.21	0.120	0.109	0.030
Quality of life	3.96 ± 0.67	3.82 ± 0.77	−0.17	0.047	3.82 ± 0.61	3.57 ± 0.75	−0.44	0.005	0.148	0.281
Health status	3.4 ± 0.82	3.46 ± 0.87	+0.22	0.116	3.13 ± 0.81	3.11 ± 0.88	−0.12	0.903	0.071	0.266
Energy	3.99 ± 0.73	3.9 ± 0.84	−0.14	0.322	3.64 ± 0.81	3.49 ± 3.83	−0.19	0.318	0.249	0.002
Autonomy for ADL	3.82 ± 0.81	3.82 ± 0.85	0.00	0.967	3.57 ± 0.81	3.44 ± 0.79	−0.25	0.372	0.136	0.058
Self-esteem	3.9 ± 0.71	3.95 ± 0.76	+0.04	0.449	3.8 ± 0.73	3.69 ± 0.77	−0.10	0.078	0.003	N. A.
Social relationships	4.12 ± 0.61	4.03 ± 3.72	−0.15	0.046	3.97 ± 0.67	3.82 ± 0.75	−0.24	0.052	0.115	0.069
Economic capacity	3.93 ± 0.81	3.89 ± 0.78	−0.06	0.795	3.81 ± 0.74	3.64 ± 0.83	−0.28	0.080	0.115	0.821
Habitat	4.29 ± 0.61	4.27 ± 0.64	−0.04	0.485	4.21 ± 0.72	4.24 ± 0.64	+0.07	0.759	0.466	0.359

*p* values were computed using general linear models (GLM) repeated measures. The results represent mean ± SD; *p* ^a^, change over time (V2 vs. V0) in nonMF at V2; *p* ^b^, change over time (V2 vs. V0) in MF at V2; *p* ^c^, group visit interaction; *p* ^d^, MF at V2 vs. nonMF at V2. Age, gender, disease duration, Hoehn&Yahr stage and LEDD at V0, and the change in LEDD from V0 to V2 were included as covariates in the model; the total score of each scale at V0 (NMSS, PDQ-39SI, and EUROHIS-QOL8) was included as covariate for the analysis of the domains. MF at V2 vs. nonMF at V2 is not applicable if test of interaction is significant (a significant test of interaction means the rates of changes over time are different between the two groups). ADL, activities of daily living; EUROHIS-QOL8, EUROHIS-QOL 8-item index; LEDD, levodopa equivalent daily dose; PDQ-39SI, Parkinson’s Disease Quality of Life Questionnaire Summary Index.

## Data Availability

The data that support the findings of this study are available from the corresponding author upon reasonable request. No computer coding was used in the completion of the current manuscript.

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
