# Peer review of "Motor Fluctuations Development Is Associated with Non-Motor Symptoms Burden Progression in Parkinson’s Disease Patients: A 2-Year Follow-Up Study"

_diagnostics, 2022, doi:10.3390/diagnostics12051147_

Round 1
Reviewer 1 Report
I think this is great work. Authors investigated the association between motor fluctuations and early non-motor burden. This issue is highly up-to-date, since clinicians must deal with levodopa-induced complications in their everyday practice.
In this article, authors bring original insight into the risk for fluctuations development and connection with the nonmotor symptoms spectrum.
I have no concerns regarding their methodology, I have just one question.
The patients who developed MF at V2 had significantly higher not only baseline LEDD but also:
- Time under levodopa therapy
- A daily dose of levodopa
- DA equivalent daily dose
Could these have an impact on the MF development?
Discussion is precisely written and conclusions are based on authors‘ results.
References are appropriate.
Author Response
Reviewer 1
I think this is great work. Authors investigated the association between motor fluctuations and early non-motor burden. This issue is highly up-to-date, since clinicians must deal with levodopa-induced complications in their everyday practice.
In this article, authors bring original insight into the risk for fluctuations development and connection with the nonmotor symptoms spectrum.
I have no concerns regarding their methodology, I have just one question.
The patients who developed MF at V2 had significantly higher not only baseline LEDD but also:
Time under levodopa therapy
A daily dose of levodopa
DA equivalent daily dose
Could these have an impact on the MF development?
Discussion is precisely written and conclusions are based on authors‘ results.
References are appropriate.
AUTHORS – Thank you very much for your comment. We agree with your comment about the importance of these variables. In the linear regression model (change in the NMSS total score from V0 to V2 as dependent variable), to develop MF was associated with an increase in the NMSS total score after adjustment to LEDD at baseline and the change in LEDD from V0 to V2, but when time under levodopa and H&Y were included as covariates, the statistical significance disappeared. As it is commented in the discussion, this indicates not only that MF development is associated with a NMS burden increase but also that those patients who developed MF are as a whole PD patients with more advanced disease. Importantly, NMS burden is related to disease progression but NMS burden at baseline was included as covariate in the model. If MF development is considered as dependent variable, time under levodopa (OR=1.018; p<0.0001), time under DA (OR=1.017; p<0.0001), daily dose of levodopa (OR=1.003; p<0.0001) and LEDD (OR=1.002; p<0.0001) were associated factors with their development but not daily dose of DA (OR=1.002; p=0.061). Although this study does not intend to identify predictive factors for the development of MF, these data suggest what is already known, that their development is partly due to a more advanced disease with a greater need for medication.
Reviewer 2 Report
Santos-Garcia and colleagues investigated the progression of NMS burden in PD patients and its association with the development of motor fluctuations. The study was performed in 330 PD patients without MF at baseline, who were recruited from 35 Spanish centers from the COPPADIS cohort and evaluated again after a 2-year follow-up. The mean NMSS total score at baseline was higher in those patients who developed MF after a 2-year follow-up. Development of MF was associated with an increase in the NMSS total score after adjusting for multiple confounders including LEDD (but the association became non-significant when controlled for H&Y score and time on levodopa treatment). The authors suggested that in PD, the development of MF is associated with a greater increase in the NMS burden after a 2-year follow-up and emphasized the importance of detection of MF and NMS.
Comments:
This is a very thoroughly conducted study examining multiple variables.
I find the article well-written and interesting.
My main comment is regarding the NMS burden and MF association becoming non-significant after taking into consideration time on Levodopa and H&Y score. MF occur earlier on Levodopa in comparison to other dopaminergic treatment, however, these are usually non-disabling and even preferred by some patients over the “OFF” state (better quality of life in some instances), therefore it is not surprising that the authors did not find the correlation when taking time on Levodopa into account. This important point should be emphasized and expanded. I feel that currently it is not given enough attention. Similarly, a short discussion about H&Y would be important.
I would suggest amending the conclusion as the authors state that the development of MF was associated with a greater NMS burden increase in the short term but do not mention the adjustment for severity and duration on levodopa.
Please specify the kind of MF disabling/moderate/mild
Table 1 I suggest including the type of test with a * beside each p-value.
If data is non-normally distributed the median should be reported.
Fig 2 I suggest adding a y-axis. Could the authors attempt to explain the right-hand side of stacked bars? What does it refer to- it is not clear to me now. Is it possible to include a legend describing this in more detail?
In Fig 2, the most important p values (pe) adjusted for multiple confounding factors are not provided. Please include the column with pe.
Fig 3 Please include a description of axis x and y
Minor comments
Please re-read the manuscript carefully for any typos.e.g., “painful associated with dystonia” (should be pain). “The mean score on all domains of the PDQ-… was the highest in patients who develop”-ed. “Time under levodopa”= time on levodopa.
In the discussion part please provide SD where means are used.
MF (early or advanced) perhaps it would be better to use disabling vs not-disabling.
Author Response
Reviewer 1
I think this is great work. Authors investigated the association between motor fluctuations and early non-motor burden. This issue is highly up-to-date, since clinicians must deal with levodopa-induced complications in their everyday practice.
In this article, authors bring original insight into the risk for fluctuations development and connection with the nonmotor symptoms spectrum.
I have no concerns regarding their methodology, I have just one question.
The patients who developed MF at V2 had significantly higher not only baseline LEDD but also:
Time under levodopa therapy
A daily dose of levodopa
DA equivalent daily dose
Could these have an impact on the MF development?
Discussion is precisely written and conclusions are based on authors‘ results.
References are appropriate.
AUTHORS – Thank you very much for your comment. We agree with your comment about the importance of these variables. In the linear regression model (change in the NMSS total score from V0 to V2 as dependent variable), to develop MF was associated with an increase in the NMSS total score after adjustment to LEDD at baseline and the change in LEDD from V0 to V2, but when time under levodopa and H&Y were included as covariates, the statistical significance disappeared. As it is commented in the discussion, this indicates not only that MF development is associated with a NMS burden increase but also that those patients who developed MF are as a whole PD patients with more advanced disease. Importantly, NMS burden is related to disease progression but NMS burden at baseline was included as covariate in the model. If MF development is considered as dependent variable, time under levodopa (OR=1.018; p<0.0001), time under DA (OR=1.017; p<0.0001), daily dose of levodopa (OR=1.003; p<0.0001) and LEDD (OR=1.002; p<0.0001) were associated factors with their development but not daily dose of DA (OR=1.002; p=0.061). Although this study does not intend to identify predictive factors for the development of MF, these data suggest what is already known, that their development is partly due to a more advanced disease with a greater need for medication.
Reviewer 2
Santos-Garcia and colleagues investigated the progression of NMS burden in PD patients and its association with the development of motor fluctuations. The study was performed in 330 PD patients without MF at baseline, who were recruited from 35 Spanish centers from the COPPADIS cohort and evaluated again after a 2-year follow-up. The mean NMSS total score at baseline was higher in those patients who developed MF after a 2-year follow-up. Development of MF was associated with an increase in the NMSS total score after adjusting for multiple confounders including LEDD (but the association became non-significant when controlled for H&Y score and time on levodopa treatment). The authors suggested that in PD, the development of MF is associated with a greater increase in the NMS burden after a 2-year follow-up and emphasized the importance of detection of MF and NMS.
Comments:
This is a very thoroughly conducted study examining multiple variables.
I find the article well-written and interesting.
My main comment is regarding the NMS burden and MF association becoming non-significant after taking into consideration time on Levodopa and H&Y score. MF occur earlier on Levodopa in comparison to other dopaminergic treatment, however, these are usually non-disabling and even preferred by some patients over the “OFF” state (better quality of life in some instances), therefore it is not surprising that the authors did not find the correlation when taking time on Levodopa into account. This important point should be emphasized and expanded. I feel that currently it is not given enough attention. Similarly, a short discussion about H&Y would be important.
AUTHORS – Thank you very much for your comment. Sorry, but I am not sure about you want to say: “MF occur earlier on Levodopa in comparison to other dopaminergic treatment, however, these are usually non-disabling and even preferred by some patients over the “OFF” state (better quality of life in some instances)”. Motor fluctuations imply that the patient develops OFF episodes and alternates with ON episodes. It is true that at the beginning they can be mild symptoms during OFF (motor and non-motor). What does happen is that patients sometimes develop mild to moderate dyskinesias, and many times they prefer to be with dyskinesias than in the OFF state. We agree with you in the fact that time under levodopa is an associated factor with MF development and expanded in the discussion the comments about this and H&Y: “The same happened when time under levodopa therapy was included as covariate in the model. It is well known that both aspects are related to the development of MF [30,31]. A more advanced H&Y stage in relation to a greater degree of denervation of the striatal nucleus and more sensitive to the development of MF [43]. On the other hand, a longer time under levodopa could imply a longer disease duration but also more time exposed to certain causative mechanisms (presynaptic and postsynaptic changes and pharmacokinetic and pharmacodynamic factors) [30,31]”.
I would suggest amending the conclusion as the authors state that the development of MF was associated with a greater NMS burden increase in the short term but do not mention the adjustment for severity and duration on levodopa.
AUTHORS – Thank you very much for your comment. We change the sentence: “In conclusion, we demonstrated for the first time in a prospective study that in PD the development of MF is associated with a greater NMS burden increase in the short-term. In practice, it is essential to detect MF early and ask about NMS, especially in patients with a greater disease severity and a longer time on levodopa”.
Please specify the kind of MF disabling/moderate/mild
AUTHORS – Thank you very much for your comment. As we explained in methods, MF were defined according to the UPDRS-IV-item 39, as a whole, but we didn´t identified how disabling MF were. It is an important point because in practice it is not only of great importance how many hours the patient spends during the OFF state but also how of severe are OFF episodes. However and unlike dyskinesia (disabling or not disabling), MF are not usually classified as disabling/moderate/mild in clinical practice. Unfortunately, we didn´t applied this classification.
Table 1 I suggest including the type of test with a * beside each p-value. If data is non-normally distributed the median should be reported.
AUTHORS – Thank you very much for your comment. Only two test were applied, Chi-square and Mann-Whitney-Wilcoxon test for discontinuous (expressed as a percentage) and continuous variables (expressed as a mean ± SD), respectively, so it is not necessary and to add a symbol can be more confusing. One-sample Kolmogorov-Smirnov test was used for determining the distribution of variables and statistical analysis were applied according to it. However, data for continuous variables was expressed as mean ± DS (as in more than 20 publications about this cohort: https://pubmed.ncbi.nlm.nih.gov/?term=coppadis&sort=date&size=200). In table 1, the standard deviation between both groups is similar.
Fig 2 I suggest adding a y-axis. Could the authors attempt to explain the right-hand side of stacked bars? What does it refer to- it is not clear to me now. Is it possible to include a legend describing this in more detail?
AUTHORS – Thank you very much for your comment. You are right. We add the symbol % in the y-axis.
In Fig 2, the most important p values (pe) adjusted for multiple confounding factors are not provided. Please include the column with pe.
AUTHORS – Thank you very much for your comment. The p value of each analysis is shown in the legend. With this figure we want to provide information about the percentage of patients with mild (NMSS 1-20), moderate (NMSS 21-40), severe (NMSS 41-70) and very severe (NMSS > 70) NMS burden at V0 and at V2 in both groups, but not including confounding factors. The analysis in which we included cofounding factors was considering the NMSS as a quantitative variable, as it was shown in Results.
Fig 3 Please include a description of axis x and y.
AUTHORS – Thank you very much for your comment. We add this comment in the legend: “QoL (PDQ-39SI) (y-axis) at baseline (V0) and after a 2-year follow-up (V2) (y-axis) in PD patients who developed MF at V2 (MF at V2 [PD-MFV2]; N=91) and those patients who didn´t developed MF at V2 (nonMF at V2 [PD-nonMFV2]; N=239)”.
Minor comments
Please re-read the manuscript carefully for any typos.e.g., “painful associated with dystonia” (should be pain). “The mean score on all domains of the PDQ-… was the highest in patients who develop”-ed. “Time under levodopa”= time on levodopa.
AUTHORS – Thank you very much for your comment. The manuscript has been reviewed and typos corrected.
In the discussion part please provide SD where means are used.
AUTHORS – Thank you very much for your comment. We added it.
MF (early or advanced) perhaps it would be better to use disabling vs not-disabling.
AUTHORS – Thank you very much for your comment. The concept of disabling or not disabling is used for dyskinesia but not for MF and OFF episodes. Some forms of classifying MF are “early or late” or “predictable or unpredictable”, but in general they generate disability. In some analysis, patients are defined as early fluctuators (e.g., https://www.sciencedirect.com/science/article/abs/pii/S1474442213701510).